# The Effect of Stretching on the Crystal Structure and Crystal Orientation of PA510/SiO_2_ Films

**DOI:** 10.3390/ma14040705

**Published:** 2021-02-03

**Authors:** Lingna Cui, Yu Dong, Yuejun Liu, Shuhong Fan

**Affiliations:** Key Laboratory of Advanced Packaging Materials and Technology of Hunan Province, School of Packaging and Materials Engineering, Hunan University of Technology, Zhuzhou 412007, China; lncui1102@126.com (L.C.); 15573335769@163.com (Y.D.); f1s1h1@163.com (S.F.)

**Keywords:** polyamide 510, X-ray diffraction, orientation, mechanical properties, optical properties, barrier properties

## Abstract

In order to explore the relationship between the microstructure and macroscopic properties of PA510/SiO_2_ films, the effect of the stretching on the crystal structure and crystal orientation of stretched PA510/SiO_2_ films was studied. It could be seen from the transmission electron microscopy (TEM) graphs that the layered SiO_2_ molecules were mainly oriented toward the machine direction (MD) and the dispersion could be improved by stretching. Through wide-angle X-ray scattering (WAXS) analysis, PA510/SiO_2_ stretched films only contained a γ crystal form. During uniaxial stretching, especially for 1 × 3 film, the γ_1_(100) crystal form was obviously oriented in the equatorial direction, and the orientation of γ_2_(004) and γ_3_(006) crystal forms could be observed in the meridian direction. According to the Herman orientation function, the orientation of the *b*-axis in the MD increased with the increase of the stretching ratio. It was worth noting that the orientation of the crystal region was more obvious. The addition of SiO_2_ and the orientation of the crystalline and amorphous regions could improve the barrier properties of the films. The changes in the optical properties of stretched films were affected by the dispersion state of SiO_2_ and the surface roughness.

## 1. Introduction

Biaxially oriented polyamide (BOPA) film has the characteristics of high strength, good barrier properties, strong light transmission, excellent dimensional stability and organic solvent resistance [1,2,3,4,5,6,7]. It is widely used in the light industry, agriculture, energy and resources environments and other fields. However, the conventional BOPA film has some limitations in application due to its poor water-absorption stability and sharp decline in tensile and bending strength [8]. The appearance of bio-based long-carbon-chain polyamides can make up for these defects in conventional BOPA films. In many applications, the comprehensive properties of bio-based long-carbon-chain polyamides are superior. In recent years, BOPA film has developed rapidly, and the amount of BOPA film produced has accounted for 50% of the use of polyamide [9,10]. In order to ensure the healthy and stability development of the polyamide film industry, it is very important to explore the relationship between microstructure changes and macro-properties of long-carbon-chain polyamides. The Kasai R&D team has developed the industrialization technology of bio-based polyamide 5X series products with excellent performance using a “bio-manufacturing” method. This is a more advanced bio-based polyamide, following the invention of traditional petroleum-based polyamide in 1920s.

It has been shown that the thermal properties, impact strength and tensile strength of bio-based long-carbon-chain polyamides can be effectively improved by adding a nano-filler such as nano-clay, nano-cellulose or graphene. Huang et al. prepared PA11/multilayer carbon nanotube composites by melt blending [11]. It was found that the storage modulus of PA11 was increased by 54% by adding 2 wt.% of multilayer carbon nanotubes. It should be noted that the presence of nano-filler can affect the crystal morphology, crystal structure and crystallinity of semi-crystalline polyamides [12]. Aidan et al. prepared polyamide/graphene oxide nanocomposites and found that the thermal stability of the nanocomposites could be improved [13]. The promotion of α-phase crystallite formation and a molecular weight change could also be observed. The macro-properties of polyamides are the concrete presentation of a microstructure, and changes in polyamide chain orientation, crystal structure and morphology can affect the macro-properties of polyamides. Penel et al. investigated the stability and mechanical properties of α, β and γ crystal forms in polyamide 6 by means of infrared spectroscopy of a multicomponent compound [14,15]. It was found that the toughness of the β crystal form was higher than that of the α and γ crystal forms. In the process of uniaxial stretching, the β crystal form can change into an α crystal form in the strain-hardening process [16]. However, most of the research on bio-based long-carbon-chain polyamides focuses on synthesis, thermomechanical properties, thermal stability, crystallization rate and crystallinity [17,18]. The relationship between the crystal structure change and the macroscopic properties of the bio-based polyamide biaxially stretched film is an almost untouched field of research.

The polyamide 510 used in this research is a new bio-based synthetic material independently researched and developed in China. Carbon emissions are greatly reduced in the monomer production process, and the raw materials are renewable resources, so the environmental benefits are considerable. The addition of lamellar silica can not only play the role of heterogeneous nucleation of spherical silica, but also improves the barrier properties of polyamide 510-like layered clay. The research work in this paper mainly focuses on the influence of stretching and the addition of lamellar SiO_2_ on the crystal orientation of 510/SiO_2_ films, and explores the influence of a change in microstructure on the mechanical properties, optical properties, thermal properties and barrier properties.

## 2. Experiment

### 2.1. Materials

Polyamide 510 resin in pellet form was supplied by Cathay Industrial Biotech (Shanghai, China). The shear melt viscosity of the polyamide 510 was 3.0 when it was measured with the capillary rheometer (RH7-I) (Malvern Panalytical, Malvern, UK) at 230 °C. The lamellar SiO_2_ was purchased from Ningbo Laboratory (China).

### 2.2. The Preparation of PA510/SiO_2_ Cast Films

The PA510/SiO_2_ composites were prepared by melt blending. Firstly, the PA510 and SiO_2_ (1 wt.%) were placed in the vacuum-drying oven (Guangce Instrument Technology, Dongguan, China) and dried at 80 °C for 12 h, then premixed in a high-speed mixer for 5 min. The premixed mixture was blended in a twin-screw extruder (TDS-35C, Nanjing Corey Instruments, Nanjing, China). The heating zone of the twin-screw extruder was divided into nine parts, and the corresponding processing temperature of each part was 210/220/230/230/235/235/235/240/240 °C. The extrusion speed and feeding speed were adjusted to 145 rpm and 9.5 rpm, respectively. After extrusion, the pellets were cut and dried in a blast-drying oven at 80 °C for 6 h. Then the PA510/SiO_2_ film was prepared by extrusion casting (FDHU-35, Guangzhou Putong Instrument, Guangzhou, China). The PA510/SiO_2_ composite particles were vacuum-dried at 120 °C for 8 h, and the particles were added to the extrusion casting machine. The heating zone casting machine was divided into nine parts, and the corresponding processing temperature of each part was 215/225/235/235/240/240/245/245/245 °C The screw speed, the linear speed of the casting roll and traction roller, and the tension of the winding roll were adjusted to 14 rpm, 1.5 m/min and 1.5 kg, respectively. By adjusting the thickness of die head and die lip, the film thickness was adjusted to 110 μm.

### 2.3. The Preparation of PA510/SiO_2_ Biaxially Stretched Films

The biaxial tensile testing machine was used to biaxially stretch (BS-1022, Shantou Dehua Machinery Factory, Shantou, China) the PA510/SiO_2_ cast film. The cast film was cut to 100 × 100 mm^2^, and then the film was placed in the sample loading area of the tensile testing machine. During the stretching process, the stretching rate, stretching ratio, stretching temperature, heat setting temperature and setting time were set to 150 mm/s, 1 × 1~3 × 3, 110 °C, 160 °C and 60 s, respectively.

### 2.4. Characterization of PA510/SiO_2_ Films

The dispersion of SiO_2_ in the films was analyzed by high-resolution transmission electron microscopy (TEM, Jeol-1230) (Sanyo Corporation, Kyoto, Japan) with a point resolution of 0.19 nm. The PA510/SiO_2_ films were embedded in epoxy resin and cut to 70 nm thickness using an ultrathin slicer at −160 °C. The detection process used a copper wire and the smallest aperture under a voltage of 90 kV.

Wide-angle X-ray scattering (WAXS, D8 Discover) was used to determine the crystal structure and the crystal orientation by irradiating the PA510/SiO_2_ films with Cu Kα radiation (λ = 1.54 Å, generator: 50 kV, 1 mA) at room temperature. In the WAXS test, the exposure time when testing perpendicular to the film surface was 5 min. In order to obtain the maximum diffraction intensity of the film, the film was stacked to be approximately 2 mm thick during the test.

Fourier transform infrared (FTIR) spectra were recorded on a Bruker Tensor 20 spectrometer (Bruker, Karlsruhe, Germany). The number of scans for each test pattern was 40 times and the test range was 700–4000 cm^−1^. In order to quantitatively describe the crystal orientation in the PA510/SiO_2_ films, the trichroic FTIR method was used for testing. First, the infrared light was set to be perpendicular to the surface of the film, and the infrared spectra in machine and transverse directions (MD and TD, respectively) were measured, which were recorded as *S_M_* and *S_T_*, respectively. Then, the angle between the surface of the film and the infrared light was deflected by 45°, and the infrared spectrum in the TD at this time was measured, recorded as *S_NT_*. Finally, the infrared spectrum *S_N_* perpendicular to the film could be obtained according to the following formula [19,20]:(1)$$S_{N}=\frac{S_{NT}\sqrt{\left(1-sin^{2}\varphi /n^{2}\right)}-S_{T}\left(1-sin^{2}\varphi /n^{2}\right)}{sin^{2}\varphi /n^{2}}$$
where *n* is the refractive index, and is reported equal to 1.62 for polyamides [20]. *φ* is the inclination angle, for which the value is 45°. According to *S_M_*, *S_N_* and *S_T_*, the structure spectrum *S*_0_ independent of tension could be obtained as:(2)$$S_{0}=\frac{\left(S_{M}+S_{N}+S_{T}\right)}{3}$$

The orientation degree of the specific molecular chain or crystal axis (*i*) of crystal relative to the stretching direction (*j*) could be calculated by the following formula [19]:(3)$$f_{ij}=\frac{D-1}{D+2}\;\times \;\frac{2}{3cos^{2}\emptyset -1}$$
where $D=A_{j}/A_{0}$ (the dichroic ratio), $A_{j}$ and $A_{0}$ are the absorption peak intensities, and ∅ is the angle between the crystal axis and the transition moment. In this study, ∅ = 0° [19,21].

Thermal analysis was carried out using differential scanning calorimetry (DSC, TA Q20). The temperature was increased from 30 °C to 260 °C at 10 °C/min and kept at 260 °C for 3 min to melt the remaining crystals and eliminate the thermal history. The crystallinity of the stretched films was calculated according to the following equation:(4)$$X_{c}\%=\frac{\Delta H_{m}}{\Delta H_{0}\left(1-x\right)}$$
where the *x* is the content of the SiO_2_, $\Delta H_{m}$ is the melting crystallization enthalpy obtained from the first heating scan and $\Delta H_{0}$ is the heat of fusion for 100% crystalline polyamide 510, which is taken as 190 J/g [22].

The oxygen permeability (OP) of unstretched film and stretched film was tested using a Mocon (Ox-Tran Model 2/21) oxygen permeability tester (Mocon, MN, USA) at 25 °C, 0% relative humidity and 1 atmospheric pressure according to ASTM D-3985-81 [23]. A mixture of 98% nitrogen and 2% hydrogen was used as the transmission gas, and 100% oxygen was used as the test gas. Four parallel experiments were carried out for each sample.

The water vapor transmission rate (WVTR) through the specimen in units of g·m^−2^·24·h was determined according to ASTM standard E96-80 [24] using a PERMATRAN-W 3/33 (Rigel Instruments, Shenzhen, China). Three parallel tests were performed on each specimen.

Tensile tests of both the MD and TD were carried out using a universal material testing machine (CMT 2202) (Sansi aspect technology, Shenzhen, China). The elongation at break and tensile strength of the PA510/SiO_2_ films were measured according to ASTM standard D638 [25]. Samples were stretched at the rate of 0.2/s. Five parallel tests were performed on each specimen.

The transmittance and haze of the PA510/SiO_2_ films were measured using a transmittance haze tester (WGT-S) (Precision Scientific Instruments, Shanghai, China) with GB/T 2410 [26] as the standard.

An atomic force microscope (AFM, Nano Manvs) (Bruker, Karlsruhe, Germany) was used to measure the three-dimensional texture of the PA510/SiO_2_ films’ surface in tapping mode. The force constant of the probe, vibration frequency and scanning speed range were 48 N/m, 330 kHz and 0.5~2.0 Hz, respectively.

## 3. Results and Discussion

### 3.1. The Effect of Stretching on Crystal Structure of PA510/SiO_2_ Films

Transmission electron microscopy (TEM) (Sanyo Corporation, Kyoto, Japan) was used to observe the dispersion of SiO_2_ in the PA510 matrix. The dispersion of SiO_2_ in the PA510/SiO_2_ stretched film with different stretching ratios is shown in Figure 1; the red circle marks the agglomeration of SiO_2_ particles in the PA510 matrix. It was found that the SiO_2_ in all the tested samples was more uniform, but the SiO_2_ had a certain degree of agglomeration in the 1 × 1 film. It can be seen in Figure 1b that the SiO_2_ is oriented along the stretching direction. So, the SiO_2_ was changed from a random arrangement to an oriented arrangement in the PA510 matrix. At the same time, it was also found that the agglomerates of SiO_2_ in the 1 × 3 film were significantly reduced, indicating that the stretching can help the dispersion of SiO_2_ particles in the PA510 matrix. In the study of Maryam et al., they also found that the layered filler was mainly oriented toward the stretching direction after stretching, and the dispersion and arrangement of fillers in the matrix materials could be improved by stretching [27]. The change of the dispersion state of SiO_2_ not only can affect the change of crystal structure of PA510 films, but also can affect the macroscopic properties. Next, we will introduce the influence of SiO_2_ on the crystal structure of PA510 biaxially stretched films.

The TD–MD surface 2D-WAXS patterns of the stretched PA510/SiO_2_ films are shown in Figure 2. From Villaseñor’s research, it was known that the parameters of the monoclinic unit cell were *a* = *b* = 4.79 Å, *c* = 40.5 Å and γ = 120°. [28] Therefore, compared with the conventional γ conformation, the amide group in the molecular chain was shortened by ~0.33 Å relative to the extended conformation. A model was established through the relationship between torsion angles: φ_1_ = −φ_2_ = ψ_1_ = −ψ_2_ = 115°. [28] In this model, the 2_1_/m molecular chain of PA510 was symmetric, and the values obtained agreed with the *c* parameter in the experiment. Therefore, the direction of a single hydrogen bond in γ crystal form is parallel to the *b* crystal axis in a monoclinic cell. For the unstretched PA510/SiO_2_ film (1 × 1), it exhibited two reflection rings for the γ_1_(100) and γ_2_(004) crystal forms. During the uniaxial stretching process, especially for the 1 × 3 film, the γ_1_(100) crystal form was obviously oriented in the equatorial direction. In the meridian direction, the orientation of the γ_2_(004) crystal form and the γ_3_(006) crystal form can be observed. The appearance of the γ_3_(006) new crystal form shows that the higher uniaxial stretching is beneficial to the orientation of crystal. The new crystal form is related to the heterogenous nucleation of the PA510 crystal on the surface of the layered SiO_2_, which can create another crystal population with a different orientation (this will be discussed later). For biaxially oriented films (2 × 2 and 3 × 3), there was an obvious orientation of γ_1_(100) and γ_2_(004) crystal forms.

The 1D-WAXD curves of the stretched PA510/SiO_2_ films are shown in Figure 3. It can be seen that there are two strong diffraction peaks at 2θ = 20.78° and 2θ = 8.53°, corresponding to the γ_1_(100) and γ_2_(004) crystal planes of the γ crystal form in Figure 2, respectively, which indicates that there was mainly a γ crystal form in all PA510/SiO_2_ stretched films. It was found that the position of diffraction peak corresponding to the γ_1_ crystal form gradually moved to a higher position, while the position of diffraction peak corresponding to γ_2_ crystal form almost did not change when the stretching temperature was 110 °C. It has been confirmed that both heat treatment and stretching can cause the crystal transformation of polyamide films. However, due to the special hydrogen bonding arrangement of PA510, it is difficult to generate an α crystal form in PA510 films. By analyzing the position and shape of diffraction peak corresponding to γ crystal form, it can be inferred that there may be a crystal form in the stretched PA510/SiO_2_ films. However, since this crystal form overlaps with the γ crystal form, it is difficult to visually analyze the crystal form from the WAXS patterns. So, this crystal form will be further explained by other means later. It can also be seen in Figure 3 that the γ_3_ crystal form corresponding to Figure 2 appeared at 2θ = 12.26° in the stretched PA510/SiO_2_ films (1 × 3). At the same time, a new diffraction peak at 2θ = 17.61° was observed that is defined as the γ_4_ crystal form. However, this crystal form does not appear in Figure 2. The reason may be that the diffraction peak intensity corresponding to the γ_4_ crystal form was lower, and was difficult to capture during the 2D-WAXS detection process. To further clarify the influence of the stretching on the crystal orientation of PA510/SiO_2_ films, we will discuss the changes of the orientation of crystalline and amorphous regions of the films by using FTIR.

### 3.2. The Effect of Stretching on Orientation of PA510/SiO_2_ Films Crystalline Phase

FTIR is a powerful method to measure the orientation of the crystal axis in the γ crystal population. From the WAXS analysis, it was found that the uniaxial stretching can give the film crystals an obvious orientation, so only spectral of the 1 × 1 and 1 × 3 films are given in the FTIR spectra. It can be seen in Figure 4a that the *S_M_*, *S_N_* and *S_T_* and *S*_0_ curves of the 1 × 1 film are almost same, indicating that the molecular chain of the unstretched PA510/SiO_2_ film was randomly oriented. This is consistent with the above WAXS analysis results. As shown in Figure 4b, the trichroic peak at 1208 cm^−1^ had a higher intensity in the MD spectrum than in the TD and ND spectra after stretching, indicating that the *b*-axis was mainly along the MD. This peak was related to the CH_2_ twisting vibration, and its transition moment was parallel to the hydrogen bond sheet of the γ crystal form. Therefore, the orientation function of *b*-axis in the crystal can be calculated by using Equation (5). The trichromatic peak at 726 cm^−1^ (not shown in Figure 4) was related to the out-of-plane vibration of the N–H. The transition moment of this peak was perpendicular to the hydrogen-bonded sheet of the γ crystal form, which can be used to calculate the orientation function of the *a*-axis in the crystal form. According to literature reports, the Herman orientation function of the *c*-axis in the crystal can be calculated according to orthogonal relationship [19,20]:*f*_cj_ = −*f*_bj_ − *f*_aj_(5)

The orientation function of the *a*-axis, *b*-axis and *c*-axis for the γ crystal form is shown in Figure 5. It can be seen that the *a*-axis in the unstretched film (1 × 1) had an orientation in the TD and ND, and this orientation is more obvious in the ND. However, the *a*-axis orientation is more obvious in the TD in the 1 × 3 PA510/SiO_2_ film. The reason for this phenomenon may be related to the tilt of the hydrogen-bonded sheet and the arrangement of the hydrogen-bonded sheet on the surface of the PA510/SiO_2_ film. For the orientation function of the *b*-axis, it was found that the orientation in MD increased with an increase in the stretching ratio. In 1 × 1 PA510/SiO_2_ film, the orientation of the *c*-axis in the TD and ND was more uniform, but the orientation of the *c*-axis in the TD and ND was more obvious in the TD for 1 × 3 film. Of course, all the crystal axes in the γ crystal form in the PA510/SiO_2_ film were preferentially oriented in the MD during the uniaxial stretching process.

Figure 4b shows that a trichromatic peak appeared at 938 cm^−1^ in the MD spectrum of the 1 × 3 PA510/SiO_2_ film. This peak corresponds to the C=O=NH plane vibration in the molecular chain. It has been reported in the literature that this peak mainly characterizes the α’ crystal form, which tends to be more parallel to the film surface [27]. The orientation of the α’ crystal form in the MD is more obvious after uniaxial stretching. However, the trichromatic peak did not appear in the TD and ND. In addition, the α’ crystal form was not found in the WAXS characterization. The main reason is that the α’ and γ crystal forms overlapped each other, and it was difficult to distinguish them in the WAXS test. The effect of stretching on the orientation of the amorphous region in the PA510/SiO_2_ films was studied by polarized infrared trichromatic light. The tricolor peak at 1121 cm^−1^ corresponded to the amorphous region of the film, and the orientation function of the amorphous region was determined according to the peak intensity (Figure 6). The amorphous orientation of the unstretched film (1 × 1) was very low compared with the stretched film (1 × 3), mainly because the segment motion of the amorphous region gradually increased during the stretching process. So, it can be found form Figure 6 that the 1 × 3 film in the amorphous region have more obvious orientations. As shown in Figure 4b, the FTIR spectrum of the PA510/SiO_2_ film had no obvious orientation in the TD, MD and ND at other spectral positions. In other words, the FTIR was not sensitive enough to accurately record all changes in the coordinate peak intensity in the PA510/SiO_2_ film. The special molecular chain arrangement in the PA510 film made the hydrogen bond density lower than that of PA6, so that the entanglement of the molecular chains was relatively low. It is worth nothing that the orientation of the amorphous region was much lower than the orientation of the crystalline region after stretching. The orientation of the crystalline and amorphous regions and the orientation of SiO_2_ during the stretching process can affect the macroscopic properties of the PA510/SiO_2_ films. Therefore, in the following studies, we focused on the influence of stretching on the thermal, mechanical, optical and barrier properties of the PA510/SiO_2_ films.

### 3.3. The Effect of Stretching on the Thermal Properties of PA510/SiO_2_ Films

The first heating curves of the PA510/SiO_2_ biaxially stretched films are shown in Figure 7. It can be seen that the melting point of the γ crystal form appeared on the heating curve around 215 °C. The stretching had little effect on the melting point of the PA510/SiO_2_ films. In order to further explore the influence of stretching on the thermal properties of the PA510/SiO_2_ films, Table 1 summarizes the influence of stretching on the melting point and crystallinity. It can be seen that uniaxial stretching can increase the crystallinity of the PA510/SiO_2_ films, which indicates that the stretching can induce the crystallization of the films. For biaxially stretched films, the degree of crystallinity was smaller than that of uniaxially stretched films. In addition to the effect of stretching on the increase of crystallinity, the addition of SiO_2_ particles can play a role in heterogeneous nucleation and increase the crystallinity of the PA510/SiO_2_ films.

### 3.4. The Effect of Stretching on the Optical Properties of PA510/SiO_2_ Films

The haze and transmittance of uniaxially stretched and biaxially stretched PA510/SiO_2_ films are shown in Figure 8. For uniaxially stretched films, the transmittance of the PA510/SiO_2_ films gradually decreased and the haze gradually increased as the stretching ratio increased. For the biaxially stretched film, the change trend of the PA510/SiO_2_ film’s transmittance and haze was consistent with the uniaxially stretched films, but the changes in haze and transmittance in the biaxially stretched film were smaller than that of the uniaxially stretched film. The change in optical properties of the PA510/SiO_2_ films was related to the microscopic morphology of the SiO_2_ and the surface roughness of the film. Thus, we will introduce these two aspects in detail later.

The AFM images of the surface morphology of the PA510/SiO_2_ films are shown in Figure 9. Figure 9a,a-1 show that the peaks and valleys on the surface of 1 × 1 PA510/SiO_2_ film were dense, fine and uniform in size (without extra-large peaks and valleys). With the increase of the stretching ratio, the raised peaks of the film surface are obviously higher than those of the unstretched film. At the same time, the transition between the peak and valley is clear, and there are some large convex peaks. The surface of the stretched film was rougher than that of the unstretched film when comparing the Ra values. By increasing the stretching ratio, the surface of the PA510/SiO_2_ film became rougher, and the light transmission performance was greatly decreased. Through the analysis of the optical properties, it was found that the stretching could change the surface gloss of the film. After taking pictures of the stretched PA510/SiO_2_ films with a camera (Figure 10), it was found that the film had gradually changed from light-transmitting to non-light-transmitting. Combined with Figure 8, Figure 9 and Figure 10, it was found that the decrease in the surface gloss of the PA510/SiO_2_ films was related to the roughness of the film surface. The extinction effect of light on a rough surface is mainly due to scattering. When the incident light hits a rough surface, the film surface reflection is replaced by diffuse reflection. The essence of diffuse reflection also obeys the law of reflection. However, due to the mutual interference of diffuse reflection, the light intensity of the scattering field is increased and the specular reflection intensity is greatly weakened. Therefore, the surface texture of the PA510/SiO_2_ films had a great influence on the optical properties.

The change in the optical properties of the PA510/SiO_2_ films was not only related to the surface texture of the films, but also was affected by the micro-morphology of the SiO_2_ in the film. According to the analysis of the TEM in Figure 1, it was found that in the unstretched film, some SiO_2_ molecules were distributed in an aggregated form. At this time, the size of the aggregates was smaller than the minimum wavelength of visible light of 380 nm, which had little effect on the transmission of visible light. So, the film had a higher transmittance and a lower haze. When the film was stretched, the SiO_2_ was oriented along the direction of external force, and the diameter of the aggregate was more than 380 nm, which affected the transmission of visible light and produced light color scattering. Therefore, the transmittance of the stretched films was lower. Especially for 1 × 3 PA510/SiO_2_ film, this effect was more obvious. In summary, the changes in the optical properties and surface gloss of the stretched film were affected by the dispersion state of the SiO_2_ and the surface roughness.

### 3.5. The Effect of Stretching on the Mechanical Properties of PA510/SiO_2_ Films

The elongation at break and tensile strength of PA510/SiO_2_ films at different stretching ratios are shown in Figure 11. It can be seen that the tensile strength of the PA510/SiO_2_ films increased with the increase in the stretching ratio, while the corresponding elongation at break continued to decrease in the MD. For the film with a stretching ratio of 1 × 3, the tensile strength in MD was increased by 113% over that of unstretched film (1 × 1), and the elongation at break decreased from 400.11% to 169.65%. In addition, the tensile strength and elongation at break of the uniaxially stretched films decreased in TD, but the change range was small. This is because the TD of the PA510/SiO_2_ films was fixed during the uniaxially stretching process in this experiment; this is known as “restricted uniaxially stretching.” In conventional uniaxial stretching, the molecular chain is free in the TD. Therefore, the molecular chain was obviously oriented in MD, but it was vertical in TD during the conventional uniaxial stretching process. However, the molecular chain of the restricted uniaxial stretching was fixed in TD, and its contraction along MD was inhibited. At the same time, the molecular chain was not vertically oriented in TD. So, the tensile strength could be maintained in the TD. Compared with uniaxially stretched films, the tensile strength and elongation at break of the biaxially stretched film are more balanced in MD and TD. This shows that the biaxially stretched films were isotropic. According to the FTIR analysis, uniaxial stretching made the amorphous region and the *b*-axis of γ crystal oriented along the MD, so the material could bear more stress in the MD, and the corresponding strength in the TD was reduced. In addition, the orientation of amorphous region and *b*-axis in MD was lower in the biaxially stretched film. So, the tensile strength in MD was lower than that in uniaxially stretched films, but the tensile strength in TD was higher than that of the uniaxially stretched films.

### 3.6. The Effect of Stretching on the Barrier Properties of PA510/SiO_2_ Films

The changes in the water vapor permeability (WVP) and oxygen permeability (OP) of the PA510/SiO_2_ films at different stretching ratios are shown in Figure 12. It can be seen that when the uniaxial stretching ratio was 1 × 3, the moisture and oxygen resistance properties of the PA510/SiO_2_ films were improved by 64.8% and 50.1%, respectively, compared with the 1 × 1 PA510/SiO_2_ film. When the biaxial stretching ratio was 3 × 3, the moisture and oxygen resistance properties of the PA510/SiO_2_ films were higher than those of the 1 × 1 film by nearly 78.6% and 63.9%, respectively. The results show that stretching can effectively improve the barrier properties of the PA510/SiO_2_ films. The increase in crystallinity, the decrease in the mobility of molecular chains in the amorphous region and the orientation of the crystal phase can all improve the barrier properties of the PA510/SiO_2_ films. The improvement of its barrier properties is not only related to these factors, but also affected by the arrangement of the SiO_2_ sheets. It can be seen from the TEM characterization that the SiO_2_ sheets tended to be aligned parallel to the film surface after stretching, and this tendency became more obvious after biaxial stretching.

## 4. Conclusions

The effect of stretching on the crystal structure, crystal orientation, optical properties, mechanical properties and barrier properties of biaxially stretched PA510/SiO_2_ films was studied. Stretching can improve the dispersion of SiO_2_ in the PA510 matrix and make SiO_2_ oriented along the stretching direction. During uniaxial stretching, especially for the 1 × 3 film, the γ_1_(100) crystal form was obviously oriented in the equatorial direction. The γ_2_(004) and γ_3_(006) crystal forms could be observed in the meridian direction. For the biaxial stretched film, there was no obvious orientation of γ_1_(100) and γ_2_(004) crystal forms in the equatorial and meridian directions. According to the Herman orientation function, the orientation of the *b*-axis in the γ crystal form in the MD increased with an increase in the stretching ratio. The α’ crystal form appeared at 938 cm^−1^ in the MD spectrum of the 1 × 3 film according to the FTIR analysis. Combined with WAXS and FTIR analyses, it was found that the PA510/SiO_2_ films contain the γ crystal form and a small amount of the α crystal form. The stretching and the addition of SiO_2_ could increase the crystallinity of the film, but the range of increase was small. The changes in crystallinity, surface roughness and crystal orientation transformed the film from light-transmitting to non-light-transmitting. The addition of SiO_2_ and the orientation of crystalline and amorphous regions could improve the barrier properties and mechanical properties of the films. This PA510/SiO_2_ biaxially stretched film is expected to be applied to packaging.

## Figures and Tables

**Figure 1 materials-14-00705-f001:**
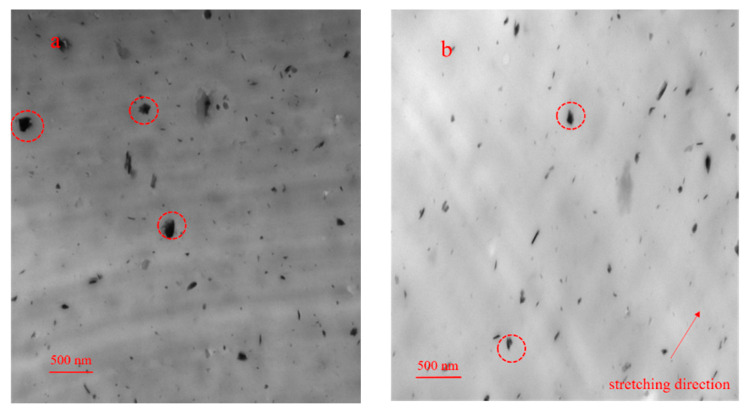
The TEM graphs of PA510/SiO_2_ films at the stretching ratio of (**a**) 1 × 1 and (**b**) 1 × 3.

**Figure 2 materials-14-00705-f002:**
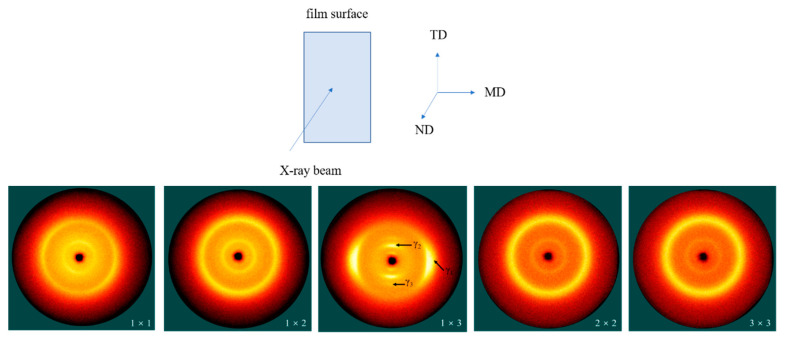
Surface (TD–MD plane) 2D-WAXS patterns for the PA510/SiO_2_ films. ND is the normal direction. And the 1 × 1, 1 × 2, 1 × 3, 2 × 2 and 3 × 3 represent the stretching ratio (TD × MD) of the films.

**Figure 3 materials-14-00705-f003:**
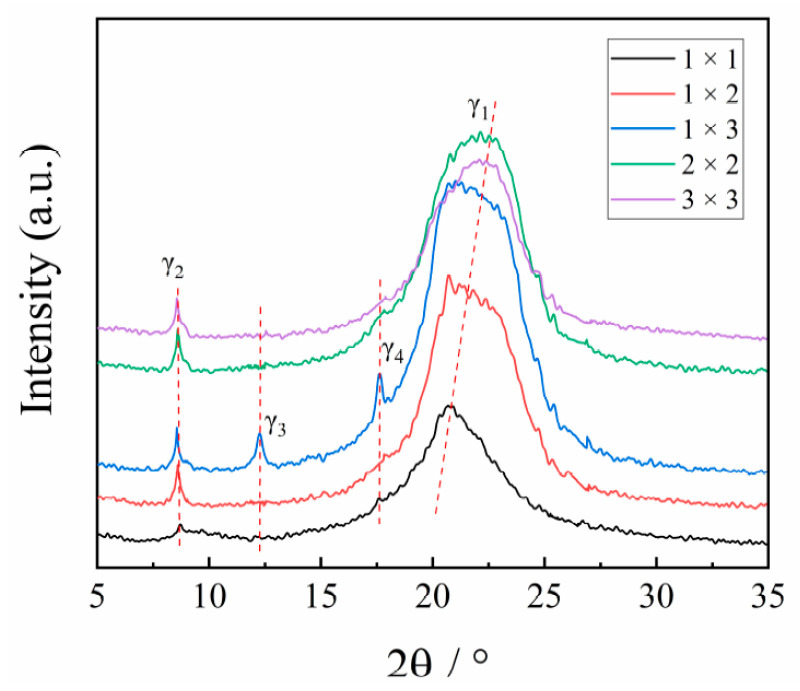
The 1D-WAXS patterns of PA510/SiO_2_ films at various stretching ratios in the surface.

**Figure 4 materials-14-00705-f004:**
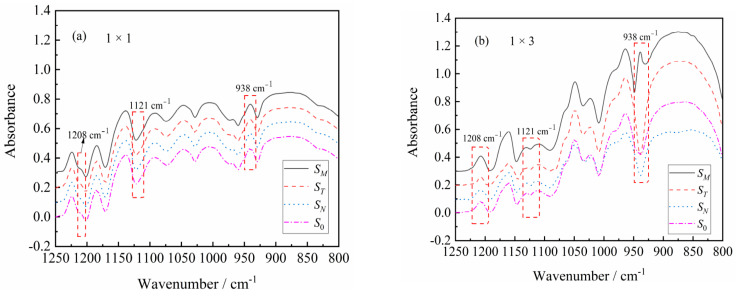
FTIR spectra of the crystal phase along the TD, MD, calculated ND and structure factor spectrum (*S*_0_) for (**a**) 1 × 1 and (**b**) 1 × 3 stretched films.

**Figure 5 materials-14-00705-f005:**
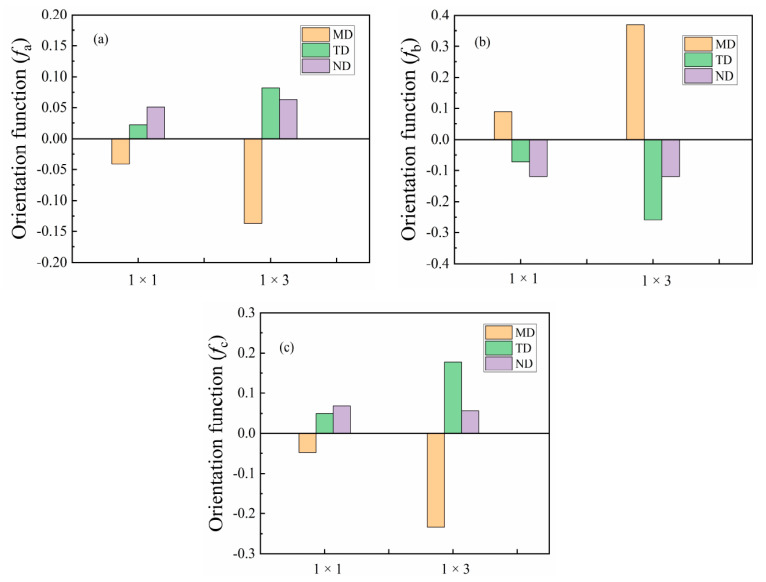
Orientation function of the γ crystal population of (**a**) *a*-axis, (**b**) *b*-axis and (**c**) *c*-axis for 1 × 1 and 1 × 3 stretched films at 1208 cm^−1^.

**Figure 6 materials-14-00705-f006:**
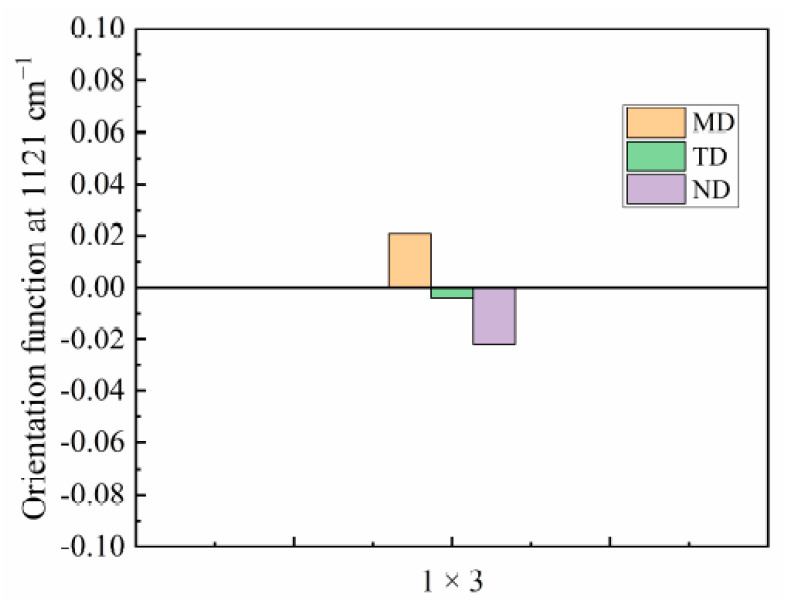
Orientation function of amorphous phase at 1121 cm^−1^ for 1 × 3 PA510/SiO_2_ films.

**Figure 7 materials-14-00705-f007:**
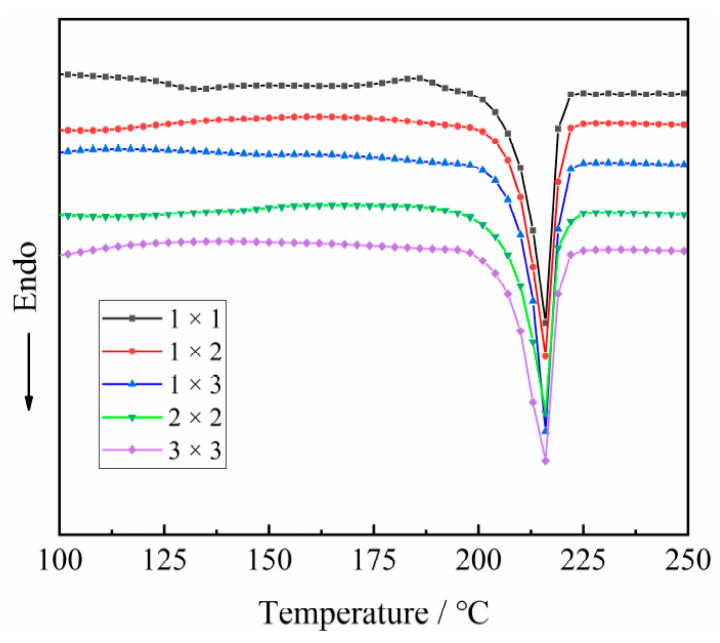
The first heating curve of the stretched PA510/SiO_2_ films.

**Figure 8 materials-14-00705-f008:**
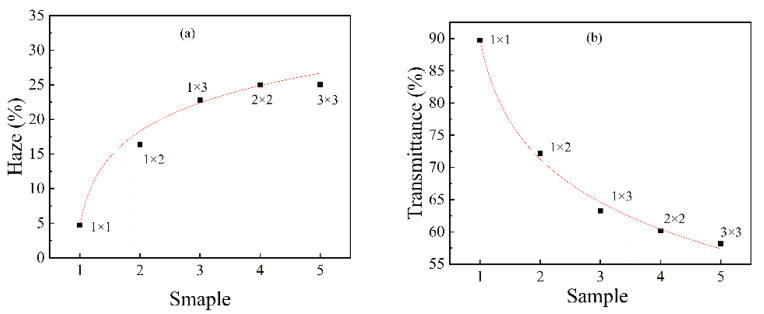
The haze and transmittance of PA510/SiO_2_ films: (**a**) haze and (**b**) transmittance.

**Figure 9 materials-14-00705-f009:**
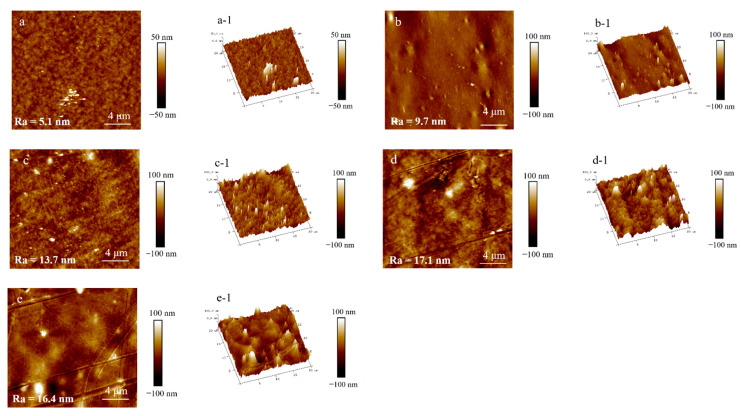
The left picture is the surface textures of PA510/SiO_2_ films, and the right picture is the 3D topography of the corresponding film. (**a**–**e**) are the surface morphology of 1 × 1, 1 × 2, 1 × 3, 2 × 2 and 3 × 3 films, respectively, and the **a-1**, **b-1**, **c-1**, **d-1** and **e-1** are the corresponding three-dimensional patterns. Ra is the roughness of the surface.

**Figure 10 materials-14-00705-f010:**
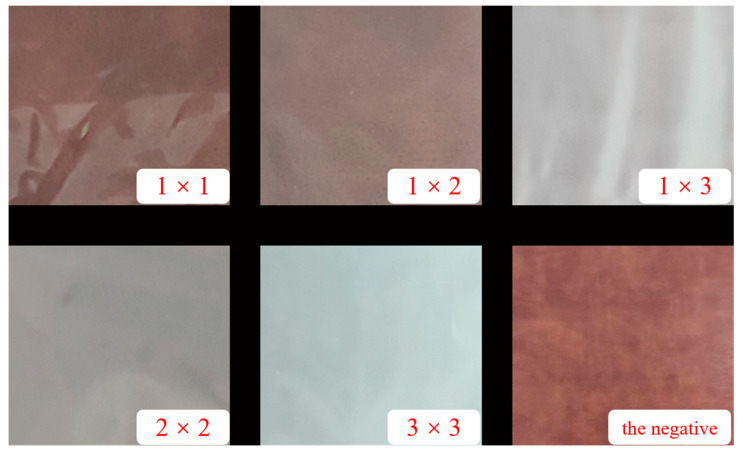
The surface pictures of PA510/SiO_2_ films with different stretching ratios; the last picture is the negative.

**Figure 11 materials-14-00705-f011:**
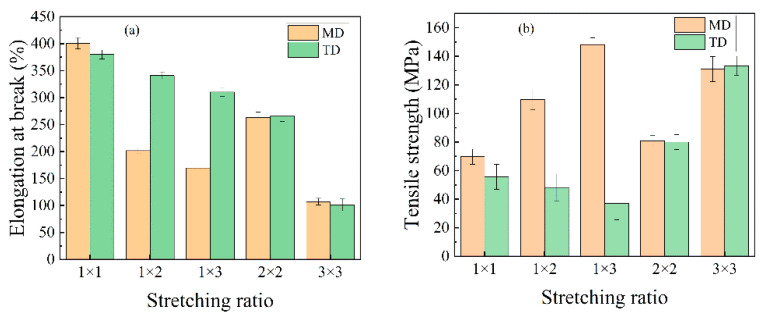
The mechanical properties of PA510/SiO_2_ films at various stretching ratios: (**a**) elongation at breaking and (**b**) tensile strength.

**Figure 12 materials-14-00705-f012:**
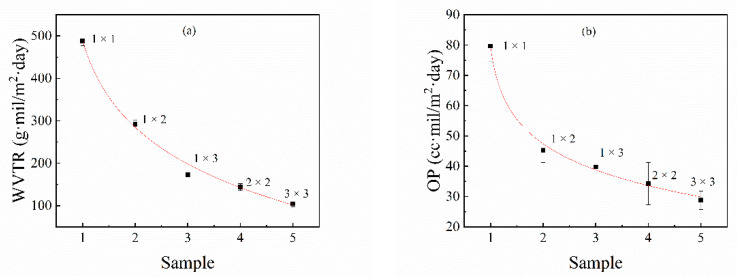
The effect on the barrier properties of PA510/SiO_2_ films: (**a**) water vapor permeability (WVP) and (**b**) oxygen permeability (OP).

**Table 1 materials-14-00705-t001:** DSC data for the first heating thermograms for PA510/SiO_2_ films.

Stretching Ratio (TD × MD)	Melting Temperature (°C)	Melting Enthalpy (J/g)	Crystallinity (%)
1 × 1	216.61	59.88	31.50
1 × 2	215.67	61.31	32.30
1 × 3	215.18	74.76	39.30
2 × 2	216.21	64.56	34.00
3 × 3	215.76	62.27	32.80

## Data Availability

The data presented in this study are available in article.

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
