# Peer review of "The Effect of Stretching on the Crystal Structure and Crystal Orientation of PA510/SiO2 Films"

_materials, 2021, doi:10.3390/ma14040705_

Round 1

Reviewer 1 Report

The authors investigate effect of stretching on the crystal structure and crystal orientation of PA510/SiO2 films, and characterization of their optical and barrier properties. The authors systematically investigate the structure and properties of stretched PA510/SiO2 films, which is interesting results, however it is hard to find a new scientific result in this manuscript for the publication.

Reviewer 2 Report

The article by Liu and co-workers details the physical properties of polyamide 510 film with interspersed SiO2. The article is well-written and details the experiments and findings clearly. It is perhaps a little verbose with some repetition of statements to enforce the core concept that the authors are making. In general it is not cumbersome to read.

The evidence that they present is supported by the analytical evidence. I would disagree that "crystal structure" should be a keyword. "X-ray diffraction" might be a better term. Crystal structure usually refers to the atomic connectivity of molecules in the solid state (either macro-molecules or small molecule/chemical compounds). The structures of the incipents in the films is not determined.

The primary concern I have is that the authors introduce the short-hand notation "MD", "TD", and "ND". Only a very brief note late in the manuscript defines MD and TD. These should be defined earlier (MD for example should be defined in the abstract where it is first used, and perhaps it should be subsequently defined again in the body text when it is first used.

The statement that MD is machine direction and TD is transverse direction appears on page 4 of the manuscript, however their abbreviations appear in the abstract (for MD) and on page 3 for MD and TD. I may have missed the notation for ND. Futhermore, throughout the manuscript the authors state "MD direction". If this statement is written out in full it would be "machine direction direction", clearly this makes no sense. As far as I can tell there is no definition for "ND". MD might also be confused with meridinal direction that is sometimes stated as well. This should be remedied to create less confusion for the reader. This is not my field of expertise, thus I ask if these terms are commonplace in the field and well-understood or if some further definition of these "directions" be provided.

Throughout the authors begin many sentences with "And". This is not correct English. It is generally better to remove the leading "And". Likewise there are a number of instances where the sentence commences with "The" which is unnecessary. Not all sentences, though there are a number that do.

Because I am not an expert in the field, the extrusion temperature and rate information on page 2 felt disconnected. Several of the temperatures were repeated or held, it is unclear how the two modes tie together.

In general this is an acceptable manuscript and I see no other additional experiments to be performed to elaborate on the findings.

Below are suggested considerations and corrections to the manuscript.

=============

Throughout: "et al" should be "et al." (note the period). "et al." should be in italics.

pg. 1 line 12: And during ... --> During ...

pg. 1 line 16: this sentence is unclear. I do not know what the authors are trying to state with "... the orientation of the amorphous region was much lower than that of the crystal region." This makes no sense. Amorphous material has no orientation (by definition). Please rephrase.

pg. 1 line 28: ... stability sharp decline ... --> ... stability and sharp decline ...

pg. 1 line 33/34 ... healthy and stability development of polyamide film ... --> ... healthy and stable development of the polyamide film ...

pg. 1 line 48: And the promotion ... --> The promotion ...

pg.2 line 49 ... change also could be observed. --> ... change could also be observed.

pg.2 line 57 chains --> chain

pg. 2 line 58 The research ... --> Research

pg.2 line 60 ... biaxially stretched film is almost in a blank area. --> ... biaxially-stretched film is an almost untouched field of research.

pg. 2 line 61: The polyamide ... --> Polyamide

pg. 2 line 65 improve --> improves

pg. 2 line 80 & 87: these temperatures are confusing to follow in correlation with extrusion speed.

pg. 3 line 100: films --> film

pg. 3 line 102: Using the wide-angle X-ray scattering (WAXS, D8 Discover) to perform the crystal structure and the crystal orientation ... --> Wide-angle X-ray scattering (WAXS, D8 Discover) was used to determine the crystal orientation ...

pg. 3 line 107/108. The sentence describing the generator settings is unnecessary. It could just be included after the wavelength in line 104; i.e. (l = 1.54A, Generator: 50 kV, 1 mA)

pg. 3 line 119: n was the --> n is the pg. 3 line 125: And te (phi) was the angle ... --> The angle (phi), is the angle ...

pg. 4 line 157: The TEM is used ... --> Transmission Electron Microscopy (TEM) was used ...

pg. 5 line 192: The final sentence in the paragraph makes no sense. Please rephrase. I do not know what the authors are trying to say to suggest better phrasing.

pg. 5 line 197: And it can ... --> It can ...

pg. 5 line 202: ... difficult to form a (alpha) crystal form in ... --> ... difficult to generate an (alpha) crystal form in ...

pg. 6 line 223: random oriented. --> randomly oriented.

pg. 6 Fig. 4: clarify identification of the 1x1 and 1x3 FTIR spectra (i.e. with (a) and (b) labels on the two charts. This is especially important because Figure 4(b) is referenced in the text, but is not explicitly stated in the Figure.

pg. 7 Fig. 5: as above. Give additional labels/clarification for each of the charts.

pg. 12 line 401: And the (gamma)2(400) ... --> The (gamma)2(400) ...

pg. 13 line 444: page numbers for reference appear incorrect, article appears to be on pg. 6 of Plastics Additives and Compounding. 2005, 7.

pg. 13 line 44: 2018 should be bold type

pg. 14 line 466: 2009 should be bold type

Author Response

Response to Reviewer 2 Comments

Point 1: The evidence that they present is supported by the analytical evidence. I would disagree that "crystal structure" should be a keyword. "X-ray diffraction" might be a better term. Crystal structure usually refers to the atomic connectivity of molecules in the solid state (either macro-molecules or small molecule/chemical compounds). The structures of the incipents in the films is not determined.

Response 1: The modification proposal is accepted and the corresponding modification has been marketed in red in the text.

Point 2: The primary concern I have is that the authors introduce the short-hand notation "MD", "TD", and "ND". Only a very brief note late in the manuscript defines MD and TD. These should be defined earlier (MD for example should be defined in the abstract where it is first used, and perhaps it should be subsequently defined again in the body text when it is first used.

Response 2: The definitions of “MD” and “TD” have been explained in the abstract and the text where they first used. The revised part has been marketed in red in the text.

Point 3: The statement that MD is machine direction and TD is transverse direction appears on page 4 of the manuscript, however their abbreviations appear in the abstract (for MD) and on page 3 for MD and TD. I may have missed the notation for ND. Futhermore, throughout the manuscript the authors state "MD direction". If this statement is written out in full it would be "machine direction direction", clearly this makes no sense. As far as I can tell there is no definition for "ND". MD might also be confused with meridinal direction that is sometimes stated as well. This should be remedied to create less confusion for the reader. This is not my field of expertise, thus I ask if these terms are commonplace in the field and well-understood or if some further definition of these "directions" be provided.

Response 3: The expression of “MD direction” in the text is not rigorous, so it has been modified and marked in red. The definition for “ND” has been explained in the text where it is first used. Generally, “MD” refers to the machine direction in stretched film. There is no abbreviation for the meridian direction. The description of direction has been modified and marked in red.

Point 4: Throughout the authors begin many sentences with "And". This is not correct English. It is generally better to remove the leading "And". Likewise there are a number of instances where the sentence commences with "The" which is unnecessary. Not all sentences, though there are a number that do.

Response 4: The modification proposal is accepted and the corresponding modification has been marketed in red in the text.

Point 5: Because I am not an expert in the field, the extrusion temperature and rate information on page 2 felt disconnected. Several of the temperatures were repeated or held, it is unclear how the two modes tie together.

Response 5: The preparation of PA510/SiO2 cast film is divided into two steps. First, the twin-screw extruder is melted and mixed, and then the cast film is prepared by the casting machine. Both machines have nine heating zones. In order to express the heating process more clearly, the description of this part in the text has been modified and marked in red.

Point 6: Throughout: "et al" should be "et al." (note the period). "et al." should be in italics.

  1. 1 line 12: And during ... --> During ...
  2. 1 line 16: this sentence is unclear. I do not know what the authors are trying to state with "... the orientation of the amorphous region was much lower than that of the crystal region." This makes no sense. Amorphous material has no orientation (by definition). Please rephrase.
  3. 1 line 28: ... stability sharp decline ... --> ... stability and sharp decline ...
  4. 1 line 33/34 ... healthy and stability development of polyamide film ... --> ... healthy and stable development of the polyamide film ...
  5. 1 line 48: And the promotion ... --> The promotion ...

pg.2 line 49 ... change also could be observed. --> ... change could also be observed.

pg.2 line 57 chains --> chain

  1. 2 line 58 The research ... --> Research

pg.2 line 60 ... biaxially stretched film is almost in a blank area. --> ... biaxially-stretched film is an almost untouched field of research.

  1. 2 line 61: The polyamide ... --> Polyamide
  2. 2 line 65 improve --> improves
  3. 2 line 80 & 87: these temperatures are confusing to follow in correlation with extrusion speed.
  4. 3 line 100: films --> film
  5. 3 line 102: Using the wide-angle X-ray scattering (WAXS, D8 Discover) to perform the crystal structure and the crystal orientation ... --> Wide-angle X-ray scattering (WAXS, D8 Discover) was used to determine the crystal orientation ...
  6. 3 line 107/108. The sentence describing the generator settings is unnecessary. It could just be included after the wavelength in line 104; i.e. (l = 1.54A, Generator: 50 kV, 1 mA)
  7. 3 line 119: n was the --> n is the pg. 3 line 125: And te (phi) was the angle ... --> The angle (phi), is the angle ...
  8. 4 line 157: The TEM is used ... --> Transmission Electron Microscopy (TEM) was used ...
  9. 5 line 192: The final sentence in the paragraph makes no sense. Please rephrase. I do not know what the authors are trying to say to suggest better phrasing.
  10. 5 line 197: And it can ... --> It can ...
  11. 5 line 202: ... difficult to form a (alpha) crystal form in ... --> ... difficult to generate an (alpha) crystal form in ...
  12. 6 line 223: random oriented. --> randomly oriented.
  13. 6 Fig. 4: clarify identification of the 1x1 and 1x3 FTIR spectra (i.e. with (a) and (b) labels on the two charts. This is especially important because Figure 4(b) is referenced in the text, but is not explicitly stated in the Figure.
  14. 7 Fig. 5: as above. Give additional labels/clarification for each of the charts.
  15. 12 line 401: And the (gamma)2(400) ... --> The (gamma)2(400) ...
  16. 13 line 44: 2018 should be bold type
  17. 14 line 466: 2009 should be bold type

Response 6: The modification proposal is accepted and the corresponding modification has been marketed in red in the text.

Point 7: pg. 13 line 444: page numbers for reference appear incorrect, article appears to be on pg. 6 of Plastics Additives and Compounding. 2005, 7.

Response 7: After checking again, the literature information given in the article is correct. The corresponding reference information is added in the additional materials.

Reviewer 3 Report

1) Introduction section must be improved with the newest findings in this field. The main task of the paper must be improved in the introduction section, because it is not clear why this paper would be interesting and whats new will be presented.

2) Abstract of the paper must be improved as it is related to the lack of the main task of the paper that is missing in the introduction section.

3) In line number 36 it is said that "and biaxial stretching film based on biological substrate". It is not clear does the film, i on the surface of the substrate or is the substrate within the film.

4) In the preparation of PA510/SiO2 cast films section description of the methods used must be improved as it is not clear that the temperatures represent the same film or the series of the films made in different temperature.

5) Figure 1 must be improved to represent findings presented in the text. It may be useful to show what areas of interest are described in the text and what changes are in those areas also described in the text.

6) As it is not clear what the main task of the paper is, because the lack of information in the introduction section. The conclusions did not represent are the planned results achieved. It must be improved with the correlation with the tasks described in the introduction section.

Author Response

Response to Reviewer 3 Comments

Point 1: 1) Introduction section must be improved with the newest findings in this field. The main task of the paper must be improved in the introduction section, because it is not clear why this paper would be interesting and whats new will be presented.

Response 1: The introduction part supplements the newest findings in this field and the main research work of the study, and the added part is marked in red.

Point 2: 2) Abstract of the paper must be improved as it is related to the lack of the main task of the paper that is missing in the introduction section.

Response 2: The abstract part has been modified and marked in red.

Point 3: In line number 36 it is said that "and biaxial stretching film based on biological substrate". It is not clear does the film, i on the surface of the substrate or is the substrate within the film.

Response 3: The sentence expression is modified and marked in red.

Point 4: In the preparation of PA510/SiO2 cast films section description of the methods used must be improved as it is not clear that the temperatures represent the same film or the series of the films made in different temperature.

Response 4: The preparation of PA510/SiO2 cast film is divided into two steps. First, the twin-screw extruder is melted and mixed, and then the cast film is prepared by the casting machine. Both machines have nine heating zones. The processing temperature of all films is the same. In order to express the heating process more clearly, the description of this part in the text has been modified and marked in red.

Point 5: Figure 1 must be improved to represent findings presented in the text. It may be useful to show what areas of interest are described in the text and what changes are in those areas also described in the text.

Response 5: The modification proposal is accepted and the corresponding modification has been marketed in red in the text.

Point 6: As it is not clear what the main task of the paper is, because the lack of information in the introduction section. The conclusions did not represent are the planned results achieved. It must be improved with the correlation with the tasks described in the introduction section.

Response 6: Through the supplement and modification of the introduction, the conclusion is modified and marked in red.

Round 2

Reviewer 1 Report

The comments about question that reviewer asked is fine. Considering with  hee comments from the other reviewers  and replies from authors, this manuscript is worth to be published.